# The Mutual Information Matrix in Hyperbolic Embedding and a Generalization Error Bound

## Abstract

Representation learning is a crucial task of deep learning, which aims to project texts and other symbolic inputs into mathematical embedding. Traditional representation learning encodes symbolic data into an Euclidean space. However, the high dimensionality of the Euclidean space used for embedding words presents considerable computational and storage challenges. Hyperbolic space has emerged as a promising alternative for word embedding, which demonstrates strong representation and generalization capacities, particularly for latent hierarchies of language data. In this paper, we analyze the Skip-Gram Negative-sampling representation learning method in hyperbolic spaces, and explore the potential relationship between the mutual information and hyperbolic embedding. Furthermore, we establish generalization error bounds for hyperbolic embedding. These bounds demonstrate the dimensional parsimony of hyperbolic space and its relationship between the generalization error and the sample size. Finally, we conduct two experiments on the Wordnet dataset and the THUNews dataset, whose results further validate our theoretical properties.

## 1 Introduction

Representation learning has gained widespread attention in the past decade as a significant task of natural language processing (NLP). Various methods have been proposed to embed words into vector spaces to facilitate further inference. The most straightforward approach is one-hot embedding, which converts each word into a binary vector corresponding to its position in the vocabulary. Latent Semantic Indexing (LSI) (Deerwester et al., 1990) generates low dimensional word embedding by applying singular value decomposition of the word-context matrix. Then a model called Latent Dirichlet Allocation (LDA) (Blei et al., 2001) was introduced based on the bag of words hypothesis. The neural language model further advanced word embeddings Bengio et al. (2000), followed by Mnih and Hinton's development of Neural Probabilistic Language Models (NPLMs) Mnih and Hinton (2008). Collobert and Weston Collobert et al. (2008) proposed pre-trained word embeddings through multitask learning. Mikolov's Word2Vec Mikolov et al. (2013) remains a powerful toolkit for training word embeddings, alongside widely used methods like GloVe Pennington et al. (2014) and FastText Bojanowski et al. (2017), which play crucial roles in various NLP tasks.

Although projecting words into an Euclidean space has achieved remarkable success in various applications, these word embedding methods need high dimensional spaces as the representing ability is positively proportional to the space dimension. Due to the immense computational and storage burden brought by high dimensions, we hope to reduce the dimensionality of the representation space. To address this issue, Nickel and Kiela applied word embedding in Poincare disk (Nickel and Kiela, 2017), benefiting from the stochastic Riemannian optimization (RSGD) (Bonnabel, 2013). This approach alleviates the dimension constraints by substituting Euclidean space with hyperbolic space. Following their work, numerous studies based on Hyperbolic embedding have been researched such as embedding graphs in Poincare ball (Sala et al., 2018). Beyond the Poincare disk, Poincare hyperplane (Ganea et al., 2018b), hyperbolic cone (Ganea et al., 2018a), and hyperbolic disks (Suzuki et al., 2019) are been considered as embedding spaces.

In this article, we aim to quantitatively assess the effectiveness of hyperbolic embedding. However, it is challenging to directly characterize the embedding error of words, because the true embedding of words in a certain space is often hard to specify accurately. To address this, we first establish a

relationship between mutual information and the hyperbolic embedding method. We then analyze the error between the embedding distance matrix and mutual information matrix, and provide theoretical properties for error bounds in hyperbolic embedding. We divide the error bounds into two components: the spatial error, which reflects the influence of the dimensions and the structure of the hyperbolic space on the embedding error; and the generalization error, which describes the relationship between the error and the sample size across different spaces. Additionally, we verify the theoretical results of hyperbolic embedding on the Wordnet and THUNews datasets.

This paper is organized as follows: Section 2 provides a brief introduction to the Lorentz model, Poincare ball, and key details of hyperbolic embedding. In section 3, we analyze the relationship between Hyperbolic embedding and the mutual information matrix for words. Then we make a further theoretical analysis for the generalization error bound of hyperbolic embedding. Section 4 outlines the experiment setups, presents and discusses the experiment results. Finally, section 5 provides a short summary of this paper.

## 2 PRELIMINARIES

We begin with a brief review of hyperbolic spaces, then give a description about Hyperbolic embedding. Next, we introduce Skip-Gram with Negative Sampling (SGNS) methods, which is adopted by the Word2Vec (Mikolov et al., 2013) and the hyperbolic embedding.

### 2.1 HYPERBOLIC SPACE

Differing from Euclidean space whose curvature is identically equal to $0$, the hyperbolic space is a smooth Riemannian manifold $\mathcal{M} = \mathbb{H}^n$ with a constant negative curvature $\kappa$, for instance, the Lorentz model is a hyperbolic space with curvature equal to $-1$.

**Definition 1.** *Let $\mathbb{R}^{(n,1)}$ denote a $(n+1)$-dimensional Minkowski space, which is a real vector space $\mathbb{R}^{n+1}$ with Minkowski dot product:*

$$\langle u, v \rangle_M := \sum_{i=1}^{n} u_i v_i - u_{n+1} v_{n+1}, \tag{1}$$

*for $u = (u_1, \cdots, u_{n+1}) \in \mathbb{R}^{(n+1)}$, $v = (v_1, \cdots, v_{n+1}) \in \mathbb{R}^{(n+1)}$ with $n \geq 2$.*

The first $n$ dimension can be viewed as a $n$-dimensional Euclidean space and company with a negative dimension. The Lorentz model is a subset of Minkowski space as one common hyperbolic model (Bridson and Haefliger, 2013).

**Definition 2.** *The Lorentz model $\mathbb{H}^{(n,1)}$ is defined as following:*

$$\mathbb{H}^{(n,1)} = \left\{ x \in \mathbb{R}^{(n,1)} \mid \langle x, x \rangle_M = -1, x_n > 0 \right\}, \tag{2}$$

where $\mathbb{H}^{n,1}$ is a smooth Riemann manifold displayed in Figure 1. The inner produce between $u \in \mathbb{H}^n$ and $v \in \mathbb{H}^n$ is defined as $[u, v] = \langle u, v \rangle_M$. The geodesic distance denotes the length of the shortest curvature on the manifold. The geodesic distance of the Lorentz model is defined as

$$d_{\mathbb{H}^{n,1}}(u, v) = \operatorname{arccosh}\left( - [u, v] \right). \tag{3}$$

The Poincare ball model in $n$-dimension is a hyperbolic space bounded in the n-dimensional sphere, which can be defined by a projection shown in Figure 1: First choose a point $P$ on Lorentz model, then form a line by extending $P$ to point $P_0 = (0, 0, \cdots, 0, -1)$, the intersection point of this line and hyperplane $\left\{ x \in \mathbb{R}^{(n,1)} : x_{n+1} = 0 \right\}$ compose a Poincare ball in $\mathbb{R}^n$. More formally, we define the Poincare ball model as follows.

**Definition 3.** *Poincare ball $(\mathbb{B}_n^c, g^B)$ is a n-dimensional smooth manifold, where*

$$\mathbb{B}_n^c = \left\{ x \in \mathbb{R}^n : c\|x\|^2 < 1 \right\}, \tag{4}$$

*Where $g^{\mathbb{B}}$ is a Riemannian metric defined as $(\lambda_x^c)^2 g^E$, $g^E = I_d$ is Euclidean metric, conformal factor $\lambda_x^c = \frac{2}{(1-c\|x\|^2)}$*

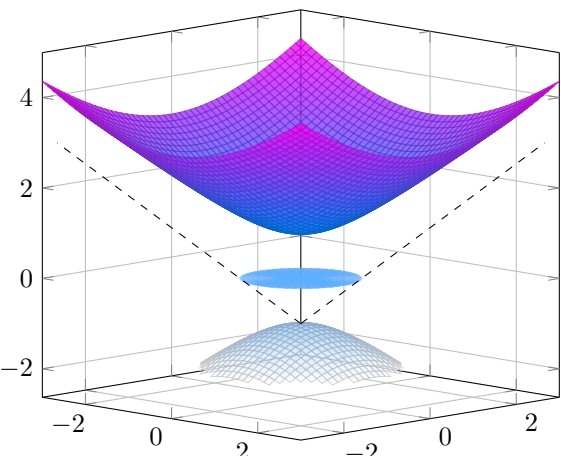

Figure 1: The Lorentz model and the Poincare ball in a 3-dimension Minkowski space

More precisely, we set the Poincare ball radius $c = 1$, then the geodesic distance between $u \in \mathbb{B}_c^n$ and $v \in \mathbb{B}_c^n$ in a Poincare ball is

$$d_{\mathbb{B}_c^n}(u, v) = \operatorname{arccosh}\left(1 + \delta(u, v)\right), \tag{5}$$

where $\delta(u, v)$ is defined as

$$\delta(u, v) = 2\frac{\|u - v\|^2}{(1 - \|u\|^2)(1 - \|v\|^2)}, \tag{6}$$

where $\|.\|$ is an Euclidean norm.

## 2.2 HYPERBOLIC EMBEDDING

Skip-gram model, first introduced in Word2Vec (Mikolov et al., 2013), is predicting the target word condition on its context which is usually the group of the words around the target word. We set the notation to follow the Omer Levy and Yoav Goldberg (Levy and Goldberg, 2014). For a word $w \in V_w$ and the context $c \in V_c$, where $V_w$ and $V_c$ are the dictionaries of words and context. A sentence of length $L$ is a bag of words $W = \{w_1, w_2, \cdots, w_{L-1}, w_L\}$. Typically, we set the contexts of word $w_i$ are the $2l$ words surrounding $w_i$ in the articles, $c = (w_{i-l}, w_{i-l+1}, \cdots, w_{i-1}, w_{i+1}, \cdots, w_{i+l-1}, w_{i+l})$. We denote the probability of $w$ condition on $c$ as $P(w \mid c)$, skip-gram model choose sigmoid function to calculate the probability in

$$P(w \mid c) = \sigma(\mathbf{w} \cdot \mathbf{c}) = \frac{1}{1 + e^{-\mathbf{w} \cdot \mathbf{c}}}, \tag{7}$$

where $\mathbf{w}, \mathbf{c} \in \mathbb{R}^d$ are the word embedding of word $w$ and context $c$, $d$ is the dimension of the embedding space.

Furthermore, Let $y$ be a sigh variable, $P(y = 1 \mid w, c)$ indicates the probability that $(w, c)$ appears in dataset $\mathcal{D}$, $P(y = 0 \mid w, c)$ indicates the probability that $(w, c)$ does not. The objective function of the skip-gram model is to maximize $P(y = 1 \mid w, c)$ for the observed date, given by $\max \log \sigma(\mathbf{w} \cdot \mathbf{c})$. Negative sampling tries to maximize $P(y = 1 \mid w, c)$ while minimum $P(y = 0 \mid w, c)$ for random negative samples, which are drawn from empirical uni-gram distribution $P_{\mathcal{D}}(c) = \frac{\#(c)}{|D|}$, where $\#(c), \#(w)$ and $\#(w, c)$ denotes the times that $c$, $w$ and $(w, c)$ appears in dataset$X$. The probability of observed data with negative samples is given by $(y = 1 \mid w, c)P(y = 0 \mid w, c_N)$, where $c_N$ is the negative sample. Then the objective function with negative samples is:

$$\max \sum_{w \in V_w} \sum_{c \in V_c} \#(w, c)\left(\log \sigma(\mathbf{w} \cdot \mathbf{c}) + k \sum_{i=1}^{n_e} \left[\log(1 - \sigma(\mathbf{w} \cdot \mathbf{c_N^i}))\right]\right), \tag{8}$$

where $n_e$ is the number of negative samples, $k$ is a factor. This objective function makes word-context pair $(w, c)$ have similar embedding. This is made by the assumption that the closer words have similar meanings.

Poincare embedding (Nickel and Kiela, 2017) changes the embedding space from Euclidean space to Poincare disk as hyperbolic space can reduce the dimension of the embedding space. The distance between points $w, c \in \mathbb{B}_d^1$ is

$$d(\mathbf{w}, \mathbf{c}) = \operatorname{arcosh}\left(1 + 2\frac{\|\mathbf{w} - \mathbf{c}\|^2}{(1 - \|\mathbf{w}\|^2)(1 - \|\mathbf{c}\|^2)}\right). \tag{9}$$

For a pair $(\mathbf{w}, \mathbf{c})$, maximum the objective function

$$\mathcal{L} = \sum_{w \in V_w} \sum_{c \in V_c} \#(w, c)\left(\log \sigma(d(\mathbf{w}, \mathbf{c})) + k\sum_{i=1}^{n_e}\left[\log(1 - \sigma(d(\mathbf{w}, \mathbf{c}_N^i)))\right]\right), \tag{10}$$

where $k$ is the negative sampling factor, $n_e$ is the number of the negative samples, $c_N$ is the negative sample drawn from distribution $P_\mathcal{D}(c) = \frac{\#(c)}{|\mathcal{D}|}$.

To optimize objective function 10, we can employ stochastic Riemannian optimization methods (Bonnabel, 2013). The embeddings of $\mathbf{w}$ and $\mathbf{c}$ are given by parametric functions $\mathbf{w} = f_\theta(w)$ and $\mathbf{c} = f_\theta(c)$, respectively. Let $\nabla_R \in \mathcal{T}_\theta\mathbb{B}$ denote the Riemannian gradient of objective function $\mathcal{L}$ at the point $\theta \in \mathbb{B}_d^1$. RSGD updates the word embedding of the form

$$\theta_{t+1} = \mathcal{R}_{\theta_t}(-\eta_t \nabla_R \mathcal{L}(\theta_t)), \tag{11}$$

where $\mathcal{R}_{\theta_t}$ denotes the retraction onto $\mathbb{B}_d^1$ at $\theta$ and $\eta_t$ denotes the learning rate at time $t$. More precisely, we give the full update form as

$$\theta_{t+1} \leftarrow \operatorname{proj}(\theta_t - \eta_t \frac{(1 - \|\theta_t\|^2)^2}{4}\nabla_E), \tag{12}$$

where $\nabla_E$ is the Euclidean gradient, $\operatorname{proj}(\cdot)$ is function to constrain the embedding within $\mathbb{B}_d^1$, which takes the follow form

$$\operatorname{proj}(\theta) = \begin{cases} \theta/\|\theta\| - \varepsilon, & \text{if } \|\theta\| \geq 1 \\ \theta, & \text{otherwise} \end{cases}. \tag{13}$$

## 3 HYPERBOLIC DISTANCE AS MUTUAL INFORMATION

The Hyperbolic space, with its infinitely nested structure, has an ultra-strong information storage capacity, which can greatly reduce the spatial dimensions required for us to seek word representations. However, the understanding of the relationship between the space dimension, sample size, and the embedding error after embedding is still lacking. As directly characterizing the truth word embedding in different spaces is quite challenging, We turn to characterizing the relationships between the embedding target of the objective function. Similar to the results from Omer Levy and Yoav Goldberg (Levy and Goldberg, 2014), SGNS in Word2Vec embeds the words and the contexts as mutual information matrix factorization: the dot product $\mathbf{w} \cdot \mathbf{c}$ equals to $PMI(x, y) = \log\frac{P(x,y)}{P(x)P(y)}$, which was obtained in the Euclidean space, we first established the relationship between hyperbolic embedding through SGNS method and mutual information matrix. we complete this analysis on Poincare embedding to figure out the relation between the hyperbolic distance matrix and mutual information matrix. Following the previous result from graph embedding(Suzuki et al., 2021) (Tabaghi and Dokmanić, 2020), we derive a bound for both the space error and the generalization error. We combine the space error and the generalization error to the embedding error and elaborate the parsimony of the embedding dimension.

### 3.1 MATRIX FACTORIZATION IN POINCARE EMBEDDING

In this part, we characterize the true values of the embedding so that we can proceed to analyze the errors in the following parts. Firstly, to characterize the error of the embedding results of hyperbolic embedding, we need to understand what the optimal solution is when using the SGNS method. To analyze the embedding from the SGNS method, we start from the optimization of the objective function 10. Then, we have

$$\mathcal{L} = \sum_{w \in V_w} \sum_{c \in V_c} \#(w, c)\log \sigma(d(\mathbf{w}, \mathbf{c})) + \sum_{w \in V_w} \sum_{\mathbf{c} \in V_c} \#(w, c)\left(k\sum_{i=1}^{n_e}\left[\log(1 - \sigma(d(\mathbf{w}, \mathbf{c}_N^i)))\right]\right). \tag{14}$$

Consider that all the negative samples are drawn from the empirical distribution, and assume that $\#(\mathbf{w})$ is sufficiently large for a large dataset and set $k = 1$,

$$\mathcal{L} = \sum_{w \in V_w} \sum_{c \in V_c} \left[ \#(w,c) \log \sigma(d(\mathbf{w},\mathbf{c})) + \frac{\#(c) \cdot \#(w) \cdot n_e}{\mid D \mid} \log\left(1 - \sigma(d(\mathbf{w},\mathbf{c}))\right) \right] \quad (15)$$

where $\mid D \mid$ is the size of the dataset. Denote $t = d(\mathbf{w},\mathbf{c})$. From the derivation of $t$,

$$\frac{\partial \mathcal{L}}{\partial t} = \frac{e^{-t}}{1 + e^{-t}} \left( \#(w,c) + \frac{\#(c) \cdot \#(w) \cdot n_e}{\mid D \mid} \right) - \frac{\#(c) \cdot \#(w) \cdot n_e}{\mid D \mid} \quad (16)$$

We found the relationship of the function d for the target embedding.

$$t = \log \frac{\#(w) \cdot \#(c)}{\mid D \mid \#(w,c)} + \log n_e. \quad (17)$$

Notice that $-\log \frac{\#(w) \cdot \#(c)}{|D|\#(w,c)}$ is the point-wise mutual information (PMI) for $(w,c)$. This result indicates that $d(\mathbf{w},\mathbf{c})$ is the mutual information between $w$ and $c$. Then we capture the truth target of the function $d$ between word $w$ and context $c$ using the SGNS method. The detailed derivation process has been placed in the appendix.

Following this result, the distance matrix between the words and the context indicates the true mutual information matrix $\boldsymbol{t} \in \mathbb{R}^{|\mathcal{V}_w| \times |\mathcal{V}_\lrcorner|}$ on the word-context pair set $\mathcal{V}_w \times \mathcal{V}_c$, the element $t_{w,c}$ of $\boldsymbol{t}$ denotes the true mutual information between word $w$ and context $c$. Each row of the mutual information matrix means the mutual information relation between a word and a context. This is equivalent to representing the word through the probability distribution of its contributions to the context. When the co-occurrence probabilities of two words with the context are similar, it also fully indicates the interchangeability between the two words, that is, the similarity between the two words.

Further, we studied the relationship between the word PMI and Pearson correlation coefficients. Interestingly, when we choose the random variables $X_w$ and $X_c$ to represent the indicator functions of whether the word and the context appear. Then $X_w X_c$ represents whether $w$ and $c$ appear simultaneously. The Pearson correlation coefficient between $X_w$ and $X_c$ is

$$Cor(X_w, X_c) = \frac{\mathbb{E}\left(X_w X_c\right)}{\sigma_w \sigma_c} = \frac{P(w,c)}{\sqrt{P(w)(1 - P(w))P(c)(1 - P(c))}}, \quad (18)$$

where $P(w,c)$ is the probability of $X_w X_c = 1$, $P(w)$ is the probability of $X_w = 1$ and $P(c)$ is the probability of $X_c = 1$. Furthermore, when we take the negative logarithm of the correlation, we discover the relationship

$$-\log Cor(X_w, X_c) = \log \frac{P(w)P(c)}{P(w,c)} + \frac{1}{2} \log(\frac{1}{P(w)} - 1)(\frac{1}{P(c)} - 1) \quad (19)$$

From the above formula, we can see that there is a strong similarity between the Pearson correlation coefficient of $X_w$ and $X_c$ and the PMI matrix. This can help us further understand the PMI matrix.

## 3.2 PARSIMONY EMBEDDING BY USING POINCARE DISK

In this section, we will introduce the parsimony property of hyperbolic embedding, which indicates that hyperbolic space can store information using a much smaller embedding space dimension compared to Euclidean space. However, our research also finds that the parsimony property of hyperbolic space requires training with more samples and has greater computational complexity when calculating distances for the encoding.

Following the results from the previous section, the distance matrix of word embedding obtained by the SGNS method in the hyperbolic embedding is a characterization of the original mutual information matrix between words. By analyzing the error between the distance matrix obtained from the embedding in hyperbolic space and the original PMI matrix, we connect the problem of word embedding to graph embedding, thereby enabling theoretical analysis of the errors in the hyperbolic embedding. We divide the obtained error bound into two parts. The first part characterizes the error

caused by the encoding space on the encoding, and the second part analyzes the error introduced by the training of the SGNS encoding method.

Considering that the variety of contexts increases exponentially with the size of the context window, and for a matrix of size $V_w \times V_c$, its rank is limited by $V_w$. In our subsequent analysis, we choose the size of the context window to be 1, that is, $V_c = V_w = V$, and at this time, the PMI matrix is a square matrix, which is easier to analyze. Under this circumstance, the embedding is obtained through the SGNS method using the word pairs.

If we focus on the the mutual information matrix $\hat{t} \in \mathbb{R}^{|\mathcal{V}| \times |\mathcal{V}|}$, the element $\hat{t}_{w,c}$ of $\hat{t}$ denotes the embedding distance between word $w$ and context $c$. In particular, we need to assume the mutual information between the same word is zero, which means the recurring words in the same sentence do not provide additional information. We denote the dataset $\mathcal{D}$ with negative data as an augmented dataset. And arguments $d(\boldsymbol{w}, \boldsymbol{c})$ of objective function $\mathcal{L}$ switch to the corresponding elements $t_{w,c}$ in matrix $\boldsymbol{t}$. Then we define a group of distance matrix called permissible matrix set $\mathcal{P}_t$ of $t$.

**Definition 4.** *Permissible matrix set in Hyperbolic space is defined as $\mathcal{P}_t = \{t_{i,j}\}$, where $t_{i,j} = d(x_i, x_j)$ and $x_i$ is a point in hyperbolic space for all $i$.*

This definition contains all distance matrices $\mathbf{t} = \{t_{i,j}\}$ that can be encoded in the hyperbolic spaces. In another way, $\hat{t}$ can be defined as

$$\hat{\boldsymbol{t}} := \underset{\mathbf{t} \in \mathcal{P}_t}{\operatorname{argmin}} \mathcal{L}_{\mathcal{D}}(\boldsymbol{t}), \tag{20}$$

where the $\mathcal{L}_{\mathcal{D}}$ is the objective function following section 2 based on the sample set $\mathcal{D}$ with negative samples. This reconstructed information matrix is estimated from the embedding result by calculating the embedding distance matrix.

Furthermore, we define the total error $\mathcal{E}$ by the difference between the objective function on the reconstructed information matrix and the real information matrix as following

$$\mathcal{E} = \mathcal{L}(\hat{\boldsymbol{t}}) - \mathcal{L}^*(\boldsymbol{t}), \tag{21}$$

where $\mathcal{L} = \mathbb{E}(\mathcal{L}_{\mathcal{D}})$ denote the expectation of loss function $\mathcal{L}_{\mathcal{D}}$.

Before beginning further analysis, we give some assumptions first.

**Assumption 1.** *For $\forall w, c \in \mathcal{V}$, $t_{w,c}$ is bounded by $\rho > 0$.*

**Assumption 2.** *The loss function $\mathcal{L}$ is Lipschitz continuous with Lipschitz constant $l$ and the absolute value of Function $\mathcal{L}$ is bounded by a constant $c$.*

The assumption 1 is a natural assumption, as in most cases, words are embedded into a bounded space. And assumption 2 is a technique setting for theoretical analysis.

To give a more precise bound, the embedding error $\mathcal{E}$ is divided into two parts as equation 22 shows.

$$\mathcal{E} = \mathcal{L}(\boldsymbol{t}) - \mathcal{L}(\hat{\boldsymbol{t}}) \leq |\mathcal{L}((\boldsymbol{t})) - \mathcal{L}((\boldsymbol{t}^*))| + |\mathcal{L}(\hat{\boldsymbol{t}}) - \mathcal{L}(\boldsymbol{t}^*)|, \tag{22}$$

where the expected minimize $\boldsymbol{t}^*$ is defined as

$$\boldsymbol{t}^* := \underset{\mathbf{t} \in \mathcal{P}_h}{\operatorname{argmin}} \mathcal{L}(\boldsymbol{t}). \tag{23}$$

Define the first part as $\mathcal{E}_1$ and the second part as $\mathcal{E}_2$, that is,

$$\mathcal{E}_1 := |\mathcal{L}(\boldsymbol{t}) - \mathcal{L}(\boldsymbol{t}^*)|, \mathcal{E}_2 := |\mathcal{L}(\hat{\boldsymbol{t}}) - \mathcal{L}(\boldsymbol{t}^*)|. \tag{24}$$

The first part $\mathcal{E}_1$ comes from the embedding ability of the embedding space called space error, the second part $\mathcal{E}_2$ comes from the sample sets called generalization error.

We first give a definition of the Gramian matrix of Poincare ball $G_p$ in equation 25,

$$G_p = \{g_{i,j}\} = \{1 + 2 \frac{||v_i - v_j||^2}{(1 - ||v_i||^2)(1 - ||v_j||^2)}\}. \tag{25}$$

Then the Poincare distance matrix is represented as

$$\boldsymbol{t} = \operatorname{arcosh}(G_p), \tag{26}$$

where $\operatorname{arcosh}(\cdot)$ is an element-wise function.

Considering the transformation from Poincare ball to Lorentz model is a bijection. In the following part, the discussion focuses on the Lorentz model. The Gramian matrix of Lorentz model is

$$G_l = \{g_{i,j}\} = \{-\langle v_i - v_j, v_i - v_j \rangle_M\}. \tag{27}$$

Then the Lorentz distance matrix is represented as

$$\boldsymbol{t} = \operatorname{arcosh}(G_l). \tag{28}$$

Applied the point-wise $\cosh(\cdot)$ function to the matrix in $\mathcal{P}_h$, we have the Permissible matrix set of $\mathcal{P}_{G_l}$. Then we transfer the estimation target to a processed point-wise mutual information matrix $\boldsymbol{t}'$ defined as equation

$$\boldsymbol{t}' = \cosh \boldsymbol{t} \tag{29}$$

by applying point-wise $\cosh(\cdot)$ function to the matrix $\boldsymbol{t}$.

**Theorem 1.** *Under the Assumption 1 and the Assumption 2, the embedding space error $\mathcal{E}_1$ in $\mathbb{H}^{(n,1)}$ is bounded by*

$$\mathcal{E}_1 \le l \operatorname{arcosh}\left(2 \sum_{i=1}^{|\mathcal{V}|-n} \cosh(\rho)\lambda_{\boldsymbol{t}',i}\right), \tag{30}$$

*where the $\lambda_{\boldsymbol{t}',i}$ is the eigenvalue for matrix $\boldsymbol{t}'$ sorted in ascending order, for $i \in \{1, 2, \cdots, |\mathcal{V}|\}$.*

This theorem gives a result that the space error $\mathcal{E}_1$ is decreasing while the dimension of embedding space is increasing.

Based on the Rademacher complexity of the objective function, the generalization error $\mathcal{E}_2$ is derived in the following theorem.

**Theorem 2.** *Under assumption 1 and assumption 2. For any $\delta > 0$, the sample error $\mathcal{E}_2$ is bounded with probability over $1 - \delta$ by following equation*

$$\mathcal{E}_2 \le \frac{2\omega(\rho)}{|\mathcal{D}|}l|\mathcal{V}|\left(\sqrt{2|\mathcal{D}|\nu \ln|\mathcal{V}|} + \frac{\kappa}{3}\ln|\mathcal{V}|\right) + 2c\sqrt{\frac{\ln\frac{2}{\delta}}{|\mathcal{D}|}}. \tag{31}$$

*where $\omega(\rho) = \cosh^2(\rho) + \sinh^2(\rho), \kappa = \frac{1}{2}, \nu = \frac{1}{4}$ for Lorentz model, $\omega(\rho) = (2\rho)^2, \kappa = 2, \nu = 4$ for Euclidean model.*

The proof of Theorem 2 mainly following the Rademacher complexity $\mathcal{R}_{\mathcal{D}}(h(\mathcal{P}_{G_l}))$ from (Suzuki et al., 2021) and the generalization error theory from (Bartlett and Mendelson, 2002). More detailed information is in the supplementary materials.

The Theorem 2 shows that $\mathcal{E}_2$ is limited by the dataset size $|\mathcal{D}|$, and the Theorem 1 shows that $\mathcal{E}_1$ is limited by the dimension of hyperbolic space. Both errors grow as the length of vocabulary grows. The space error aspect: The Theorem 1 explains that hyperbolic embedding, facilitated by nonlinear transformations within Minkowski space, enables the compression of high-dimensional Euclidean information into a lower-dimensional framework. Importantly, this embedding technique preserves a linear approximation to the target matrix prior to undergoing the nonlinear transformation. Since it directly analyzes the error brought by the space, it is very tight. The generalization error aspect: The Theorem 2 highlights that training within a low-dimensional hyperbolic space necessitates a larger sample size relative to that required in high-dimensional Euclidean space to ensure sufficient training efficacy. This observation underscores the interplay between sample size and the dimension of the space selected for embedding. For the Theorem 2, as it depends on the characterization of the Rademacher complexity, when there is a more delicate characterization of the complexity of hyperbolic space, this bound can also be further refined. Based on what we currently understand, the results indicate that the performance of the Theorem 2 matches our experimental results.

## 4 EXPERIMENT RESULT

To further validate our theoretical findings, we conducted the following experiments. We conduct experiments on a smaller dataset: Wordnet mammals, and a more complicated dataset: THUNews. We inspected the dimension of the Gramian matrix and the reconstructed distance matrix, and we also tested the embedding result in different sample sizes. All these experiments are run on a MacBook Pro with M1 chips.

## 4.1 HYPERBOLIC EMBEDDING TEST ON WORDNET MAMMALS

Wordnet is a classical word embedding dataset, which was free to use on https://wordnet.princeton.edu. Wordnet is a large lexical English database. Nouns, verbs, adjectives, and adverbs are grouped into sets of cognitive synonyms (synsets), each expressing a distinct concept. Considering the computation consumption, we applied an embedding algorithm on the mammal subset which is combined with different mammal nouns. This dataset has 1180 words and 6541 word pairs.

From the result of Section 3, we can approximate the point-wise mutual information matrix by constructing the distance matrix between words. From the information theory perspective, there is more information in the mutual information matrix as the mutual information matrix has a higher rank. To investigate the embedding ability of different embedding spaces, we compare the restored point-wise mutual information matrix of different spaces.

Table 1: The rank of embedding distance matrices in the **Euclidean** spaces of different dimensions trained on 1180 words by SGNS method

| Dimension | 10 | 100 | 200 | 300 | 800 | 1000 |
|---|---|---|---|---|---|---|
| Distance | 9 | 72 | 126 | 166 | 294 | 327 |
| Dot | 2 | 3 | 5 | 6 | 9 | 10 |

NOTES: Dimension refers to the dimension of the Euclidean spaces. Distance denotes $d(\boldsymbol{w}, \boldsymbol{c}) = ||\boldsymbol{w} - \boldsymbol{c}||_2$, which is the distance in the Euclidean space. Dot denotes $d(\boldsymbol{w}, \boldsymbol{c}) = \boldsymbol{w} \cdot \boldsymbol{c}$, which is the dot product. The higher the rank of the matrix, the more information the embedding distance matrices contain.

Table 2: The rank of embedding distance matrices and Gramian matrices in the **Poincare** spaces of different dimensions trained on 1180 words by SGNS method

| Dimension | 2 | 4 | 6 | 8 | 10 |
|---|---|---|---|---|---|
| Distance | 544 | 644 | 689 | 687 | 686 |
| Gramian | 3 | 4 | 5 | 5 | 6 |

NOTES: Dimension refers to the dimension of the Poincare spaces. Distance denotes $d(\boldsymbol{w}, \boldsymbol{c})$, which is defined in (5), the distance in Poincare space. Gramian refers to the gramian matrix in the Poincare spaces defined as (25). We can observe that the low-dimensional Poincare space can store information from high-dimensional PMI matrices, and the dimensions of their Gramian matrices, as shown in Theorem 1, are low-dimensional.

We test the rank of the restored point-wise mutual information matrix which is in the shape of $1180 \times 1180$. To test the rank of the matrix, we sum from large eigenvalues to small ones until the summation reaches 90% of the trace of the matrix. The number of eigenvalues count is the rank of the matrix. From Table 1, we test $d(\mathbf{w}, \mathbf{c})$ measured by distance and dot product. In Table 2, we test the rank of the restored point-wise mutual information matrix and the Gramian matrix defined as equation 29. It is obvious that even the rank of the distance matrix in the 2-dimension Poincare ball is much higher than in Euclidean spaces, which means that the Poincare ball preserves much more super-linear information. Furthermore, we give the rank of the distance matrix and the Gramian matrix $\boldsymbol{t}'$ in Table 3.

The result of the Lorentz model leads to the result of Theorem 1, which indicates that Hyperbolic space contains more super-liner information, but liner in the Gramian matrix. This result means that we can conserve dimensions of the encoding space through the use of hyperbolic surfaces.

To test the result of Theorem 2, we train hyperbolic embedding in the Lorentz model of all nouns in Wordnet, which has 15500 words and 743087 word pairs. Due to the computational constraint, we only test in a 2-dimensional Lorentz model, the training loss is 1.39762, which is higher than the Mammols dataset as theorem 2 shows. we compared the error changes of word embedding on the WordNet dataset under different training sample sizes and found that the training error in hyperbolic

Table 3: The training loss, the rank of embedding distance matrices and Granian matrices in **Lorentz** space of different dimensions trained on 1180 words by SGNS method

| Dimension | 2 | 4 | 6 | 8 | 10 |
|---|---|---|---|---|---|
| Distance | 13 | 649 | 660 | 666 | 666 |
| Gramian | 4 | 4 | 5 | 5 | 6 |
| Training Loss | 0.69868 | 0.94085 | 0.94228 | 0.94506 | 0.94465 |

NOTES: Dimension refers to the dimension of the Lorentz spaces. Distance denotes $d(\boldsymbol{w}, \boldsymbol{c})$, which is defined in (3). Gramian refers to the gramian matrix in Lorentz space defined as (28). The rank of the distance matrix in Lorentz space is slightly less than that in the Poincare space. The dimensions of their Gramian matrices, as shown in Theorem 1, are low-dimensional.

space decreases more slowly with the increase of sample size compared to Euclidean space. Moreover, hyperbolic space requires more than 70,000 samples to achieve to achieve convergence of the training error. While Euclidean space only needs 50,000 samples to achieve convergence of the training error. This is consistent with the theorem's prediction that hyperbolic space requires a larger sample size for training.

### 4.2 HYPERBOLIC EMBEDDING TEST ON THUNEWS

We further conducted experiments on our theoretical results using the Thucnews dataset. The Thucnews dataset contains 65000 Chinese news across 10 categories. We extracted 27 million word pairs from the news in Thucnews and embedded all 1496 words using the SGNS method in both hyperbolic and Euclidean spaces. The experimental results are as follows:

1, The reconstructed high-dimensional distance matrix demonstrates the powerful dimensionality compression capability of hyperbolic space. Since the Poincare surface of d dimensions is projected from the $(d+1)$-dimensional Lorentz model, its Gramian matrix, as revealed in our Theorem 1, is indeed of $(d+1)$-dimensions, which also explains the linear core behind the nonlinear functions in hyperbolic space. The dimensions of the word mutual information matrix are 1,377, and after adding negative sample regularization, the dimensions are 1,387. The experimental results for the Poincare surface are in table 4:

Table 4: The rank of embedding distance matrices and Gramian matrices in **Lorentz** Space of different dimensions trained on 1496 words by SGNS method

| Dimension | 2 | 4 | 6 | 8 |
|---|---|---|---|---|
| Rank of the distance matrix | 719 | 836 | 849 | 849 |
| Rank of the Gramian matrix | 3 | 5 | 7 | 9 |

NOTES: Dimension refers to the dimension of the Lorentz spaces. $d(\boldsymbol{w}, \boldsymbol{c})$ is defined in (3), the distance in Lorentz space. Gramian refers to the gramian matrices in the Lorentz spaces defined as (28). Compared to Euclidean space, the low-dimensional Lorentz space stores more information about the PMI matrix, demonstrating the Parsimony property of hyperbolic embedding. The result of the Gramian matrix also verifies Theorem 1.

Taking into account that a higher dimensional reconstruction distance matrix in word embedding can better preserve the differentiated information between different words in the PMI matrix, it helps us to better recognize and utilize these words. The high-dimensional Euclidean space is clearly weaker in preserving the rank information in the PMI matrix compared to the low-dimensional hyperbolic space. The experimental results for the Euclidean space are in table 5:

2, The results are consistent with the findings of our Theorem 2. We can see that the embedding loss in hyperbolic space is greatly affected by the size of samples, while in Euclidean space when there is an ample amount of samples, it is less affected by changes in the sample size. To achieve the same training effect, the hyperbolic space with lower dimensions requires more samples. As the final convergence errors are close, we choose to conduct comparative experiments using a 2-dimensional Poincare space and a 400-dimensional Euclidean space. The results are presented in plot 2.

Table 5: The rank of embedding distance matrices in **Euclidean** Space of different dimensions trained on 1496 words by SGNS method

| Dimension | 100 | 200 | 400 | 600 | 800 |
|---|---|---|---|---|---|
| Rank | 67 | 72 | 187 | 244 | 294 |

NOTES: Dimension refers to the dimensions of the Euclidean spaces. Rank refers to the rank of the reconstructed distance matrices. $d(\boldsymbol{w}, \boldsymbol{c})$ is the distance in Euclidean space.

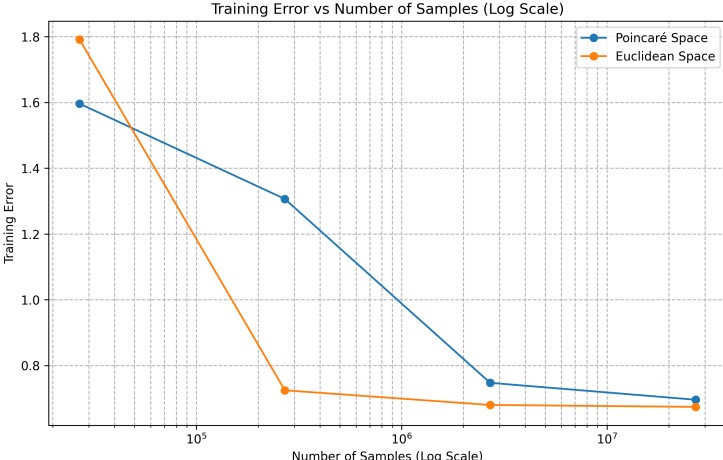

Figure 2: The plot of the number of training samples and training error in the 2-dimensional Poincare space and the 400-dimensional Euclidean space. As described in Theorem 2, compared to Euclidean space, the training error convergence in hyperbolic space requires larger samples.

3, Training in the hyperbolic space is more time-consuming. Under the same training parameters such as the number of samples, number of iterations, batch size, etc., it takes 48 seconds for a training iteration in the 400-dimensional Euclidean space, while it takes 91 seconds for a training iteration in the 2-dimensional Poincare space. The main reason for this phenomenon is that hyperbolic encoding requires the use of the RSGD (Riemannian Stochastic Gradient Descent) method, which involves additional computations for the Riemannian curvature at the current point when calculating gradients, and this curvature computation is often quite complex, leading to significant additional computational consumption. Additionally, calculating the distance between points in hyperbolic space requires the use of a complex distance function, which also significantly increases the computational complexity of the hyperbolic embedding. These experimental results tell us that the space-compression property of hyperbolic encoding requires a substantial amount of additional computational consumption and larger samples.

## 5 CONCLUSION

In this work, we give a brief analysis of the Hyperbolic embedding from the mutual information matrix. We reveal the relationship between the point-wise mutual information matrix and the distance matrix of embedding. Then following this result, we give an analysis of the errors of the hyperbolic embedding including the embedding errors and the generalization errors. Our findings indicate that low-dimensional hyperbolic spaces can accommodate more linear structures of mutual information, highlighting the equivalence between the Gramian matrix in hyperbolic embedding and the dimension of the space. Furthermore, we demonstrate that hyperbolic embedding is more unstable during training than its Euclidean counterpart, necessitating more samples for effective training. This analysis illustrates the relationship between

the dimension of embedding space, training dataset, vocabulary length, and the embedding error. These theoretical insights significantly enhance our comprehension of ongoing research initiatives,

including Hyperbolic Neural Networks (HNN) and Hyperbolic Graph Convolutional Networks (HGCN). They provide a more profound appreciation of the benefits and limitations associated with utilizing hyperbolic space for data embedding, which is crucial for advanced analysis and inferential tasks.

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

# A    APPENDIX

## A.1    THE DEDUCTION FROM THE DISTANCE MATRIX TO PMI MATRIX

To analyze the embedding from the SGNS method, we start from the optimization of the objective function 10. Then, we have

$$
\mathcal{L} = \sum_{w \in V_w} \sum_{c \in V_c} \#(w,c) \left( \log \sigma(d(\mathbf{w},\mathbf{c})) + k \sum_{i=1}^{n_e} \left[ \log(1 - \sigma(d(\mathbf{w},\mathbf{c}_N^i))) \right] \right)
$$

$$
= \sum_{w \in V_w} \sum_{c \in V_c} \#(w,c) \log \sigma(d(\mathbf{w},\mathbf{c})) + \sum_{w \in V_w} \sum_{\mathbf{c} \in V_c} \#(w,c) \left( k \sum_{i=1}^{n_e} \left[ \log(1 - \sigma(d(\mathbf{w},\mathbf{c}_N^i))) \right] \right).
$$
(32)

Notice that all the negative samples are drawn from the empirical distribution, we combine negative samples for the same word $w$,

$$
\mathcal{L} = \sum_{w \in V_w} \sum_{c \in V_c} \#(w,c) \log \sigma(d(\mathbf{w},\mathbf{c})) + \sum_{\mathbf{w} \in V_w} \left( k \sum_{i=1}^{\cdot n_e} \#(w) \left[ \log(1 - \sigma(d(\mathbf{w},\mathbf{c}_N^i))) \right] \right)
$$
(33)

Assume that $\#(\mathbf{w})$ is sufficient large for a large dataset and set $k = 1$, we have

$$
\mathcal{L} = \sum_{w \in V_w} \sum_{c \in V_c} \#(w,c) \log \sigma(d(\mathbf{w},\mathbf{c})) + \sum_{w \in V_w} \left( \#(w) n_e \cdot \mathbb{E} \left[ \log \left( 1 - \sigma(d(\mathbf{w},\mathbf{c}_N^i)) \right) \right] \right)
$$

$$
= \sum_{w \in V_w} \sum_{c \in V_c} \#(w,c) \log \sigma(d(\mathbf{w},\mathbf{c})) + \sum_{w \in V_w} \sum_{c \in V_c} \frac{\#(c) \cdot \#(w) \cdot n_e}{\mid D \mid} \left[ \log \left( 1 - \sigma(d(\mathbf{w},\mathbf{c})) \right) \right]
$$

$$
= \sum_{w \in V_w} \sum_{c \in V_c} \left[ \#(w,c) \log \sigma(d(\mathbf{w},\mathbf{c})) + \frac{\#(c) \cdot \#(w) \cdot n_e}{\mid D \mid} \log \left( 1 - \sigma(d(\mathbf{w},\mathbf{c})) \right) \right]
$$
(34)

where $\mid D \mid$ is the size of the dataset. Then we expend $\sigma(\cdot)$,

$$
\mathcal{L} = - \sum_{w \in V_w} \sum_{c \in V_c} \left[ \#(\mathbf{w},\mathbf{c}) \log \left( 1 + e^{-d(\mathbf{w},\mathbf{c})} \right) \right.
$$

$$
\left. + \frac{\#(c) \cdot \#(w) \cdot n_e}{\mid D \mid} \left( \log \left( 1 + e^{-d(\mathbf{w},\mathbf{c})} \right) - \log \left( e^{-d(\mathbf{w},\mathbf{c})} \right) \right) \right]
$$

$$
= - \sum_{w \in V_w} \sum_{c \in V_c} \left[ \left( \#(w,c) + \frac{\#(c) \cdot \#(w) \cdot n_e}{\mid D \mid} \right) \log \left( 1 + e^{-d(\mathbf{w},\mathbf{c})} \right) \right.
$$

$$
\left. + \frac{\#(c) \cdot \#(w) \cdot n_e}{\mid D \mid} d(\mathbf{w},\mathbf{c}) \right]
$$
(35)

Derived by $t$, where $t = d(\mathbf{w}, \mathbf{c})$, we have

$$\frac{\partial \mathcal{L}}{\partial t} = \frac{e^{-t}}{1 + e^{-t}} \left( \#(w, c) + \frac{\#(c) \cdot \#(w) \cdot n_e}{\mid D \mid} \right) - \frac{\#(c) \cdot \#(w) \cdot n_e}{\mid D \mid} \tag{36}$$

When reaching the maximum, we set the derivative to zero:

$$\frac{e^{-t}}{1 + e^{-t}} \left( \#(w, c) + \frac{\#(c) \cdot \#(w) \cdot n_e}{\mid D \mid} \right) = \frac{\#(c) \cdot \#(w) \cdot n_e}{\mid D \mid} \tag{37}$$

With some simplification, we have

$$t = \log \frac{\#(w) \cdot \#(c)}{\mid D \mid \#(w, c)} + \log n_e. \tag{38}$$

Notice that $-\log \frac{\#(w) \cdot \#(c)}{\mid D \mid \#(w, c)}$ is the point-wise mutual information (PMI) for $(w, c)$. This result indicates that $d(\mathbf{w}, \mathbf{c})$ is the mutual information between $w$ and $c$.

## A.2  Proof of the Theorem 1

In this subsection, we provide the details of the proof of Theorem 2.

We consider the matrix $t'$ defined in equation 29. We have the following lemma.

**Lemma 1.** *Let $t'$ be the hyperbolic Gram matrix for a set of points $x_1, \cdots, x_N \in \mathbb{H}^{(n,1)}$, Then $t' = t^+ + t^-$, and*

$$t^+, t^- \text{ is positive definite}$$
$$\text{rank } t^+ \leq n$$
$$\text{rank } t^- \leq 1$$
$$\text{diag}\{t'\} = -1$$
$$e_i^\top t' e_j \leq -1, \text{where}\{e_i\} \text{ is standard basis}$$

*Conversely, matrix $\text{arccosh}(t')$ that satisfies the above conditions is a hyperbolic distance for a set of $N$ points in $\mathbb{H}^{(n,1)}$.*

This lemma is proved in Proposition 1 of (Tabaghi and Dokmanić, 2020). Combined with the definition of $t^\star$ in equation 23, we can easily acquire the result of theorem 1.

## A.3  Proof of Theorem 2

In this subsection, we provide the details of the proof of Theorem 2.

To prove this theorem, we first give the definition of Rademacher complexity.

**Definition 5.** *For a training dataset $\mathcal{D} = \{x_1, x_2, \cdots, x_m\}$, which has $m$ samples. Rademacher complexity of a function family $\mathcal{F} = \{f \mid f : \mathcal{X} \to \mathbb{R}\}$ is defined as $\mathcal{R}_D(\mathcal{F})$*

$$\mathcal{R}_D(\mathcal{F}) = \mathbb{E}_\mathcal{D} \mathbb{E}_{\{\sigma_i\}} \left[ \frac{1}{m} \sup_{f \in \mathcal{F}} \sum_{i=1}^m \sigma_i f(x_i) \right] \tag{39}$$

*Where $\mathbb{E}_\mathcal{D}$ denotes the expectation for every sample $x_i$ under sample distribution. The $\sigma_i$, for $i \in \{1, 2, \cdots, m\}$ denoted a binary random variable in $\{-1, +1\}$ with equal probability.*

Following the results from previous work (Bartlett and Mendelson, 2002), (Kakade et al., 2008), we have theorem 3

**Theorem 3.** *Under assumption 2, For any $\delta > 0$, the following inequality holds with probability over $1 - \delta$ for all $f \in \mathcal{F}$*

$$\mathbb{E}(\mathcal{L}(f)) \leq \mathcal{L}_\mathcal{D}(f) + 2l\mathcal{R}_\mathcal{D}(\mathcal{F}) + c\sqrt{\frac{\log(1/\delta)}{2m}} \tag{40}$$

*Where $\mathcal{R}_\mathcal{D}(\mathcal{F})$ is the Redamacher complexity of a function class $\mathcal{F}$, $l$ is the Lipschitz constant of function $\mathcal{L}()$ and $c$ is the upper bound in Assumption 2.*

From theorem 3, for any $\delta > 0$, we have equation 41 with probability over $1 - \delta$ from equation 20 and equation 23.

$$\mathcal{L}\left(\hat{t}\right) - \mathcal{L}\left(t^*\right) \leq \left(\mathcal{L}_{\mathcal{D}}\left(t^*\right) - \mathcal{L}\left(t^*\right)\right) + \left(\mathcal{L}_{\mathcal{D}}\left(\hat{t}\right) - \mathcal{L}_{\mathcal{D}}\left(t^*\right)\right) + \left(\mathcal{L}\left(\hat{t}\right) - \mathcal{L}_{\mathcal{D}}\left(\hat{t}\right)\right)$$

$$\leq 2l\mathcal{R}_{\mathcal{D}}\left(h(\mathcal{P}_{\mathbf{t}})\right) + 2c\sqrt{\frac{\log\left(2/\delta\right)}{2m}} \tag{41}$$

The function family is defined by $h(\mathcal{P}_{\boldsymbol{t}})$, which is determined by distance matrix $\boldsymbol{t}$, and $h()$ is a transform function. Then we give the Rademacher complexity $\mathcal{R}_{\mathcal{D}}\left(h(\mathcal{P}_{G_l})\right)$, where the $h() = \mathrm{arcosh}()$ and distance matrix $\boldsymbol{t}$ is Gramian matrix $G_l$, in the form

$$\mathcal{R}_{\mathcal{D}}\left(h(\mathcal{P}_{G_l})\right) := \mathbb{E}_{\mathcal{D}}\mathbb{E}_{\sigma}\left[\frac{1}{m}\sup_{G_l \in \mathcal{P}_{G_l}}\sum_{i=1}^{m}\sigma_i\,\mathrm{arcosh}\left(G_l\right)\right] \tag{42}$$

**Lemma 2.** *Under assumption 1, the $\mathcal{R}_{\mathcal{D}}\left(h(\mathcal{P}_{G_l})\right)$ is bounded by following inequality:*

$$\mathcal{R}_{\mathcal{D}}\left(h(\mathcal{P}_{G_l})\right) \leq \omega\left(\rho\right)\left(\sqrt{\frac{2m\nu\ln|\mathcal{V}|}{m}} + \frac{\kappa|\mathcal{V}|\ln|\mathcal{V}|}{3m}\right) \tag{43}$$

*where $\omega(\rho) = \cosh^2(\rho) + \sinh^2(\rho), \kappa = \frac{1}{2}, \nu = \frac{1}{4}$ for Lorentz model, $\omega(\rho) = (2\rho)^2, \kappa = 2, \nu = 4$ for Euclidean model.*

The proof of this lemma is following the proof of Lemma 11 in (Suzuki et al., 2021).

The experiment code is open at https://github.com/flaneur10/Hyperbolic-embedding-error.

