# OpenReview forum: "The Mutual Information Matrix in Hyperbolic Embedding and a Generalization Error Bound"
_ICLR.cc/2025/Conference — Submitted to ICLR 2025_

### Official Review · Reviewer_Yd2w · 2024-11-01

**Soundness:** 2
**Presentation:** 1
**Contribution:** 2
**Rating:** 3
**Confidence:** 4

**Summary:**

This paper discusses the relationship between the point-wise mutual information matrix and the hyperbolic distance. Furthermore, the authors establish generalization error bounds for hyperbolic embedding. These bounds demonstrate the dimensional parsimony of hyperbolic space and its relationship between the generalization error and the sample size. Experiments on the Wordnet dataset and the THUNews dataset validate the theoretical properties.

**Strengths:**

1) Connecting hyperbolic embedding with mutual information is interesting.

**Weaknesses:**

1) The motivation of connecting hyperbolic embeddings and PMI is unclear. Both are distance measure, hyperbolic distance captures similarity of hierachies, while PMI quantifies the discrepancy between the probability. What do you mean by the “equivalence between the Gramian matrix in hyperbolic embedding and the dimension of the space”?
2) What is the research questions that you want to answer in the Experiment section? The authors said that the theoretical findings are evaluated by conducted the experiments. However, it is unclear how the experimental results related to the theoretical findings. Which theorems (theorem 1 or 2? ) you want to answer? It would be much clear if the authors list the research questions. What do you really want to evaluete and compare.
3) I could not understand what do the tables in the experiment section want to tell us? perhaps the authors want to show some correlation between dimension and mutual information matrix? then it is better to plot it with some line plots.

**Questions:**

See my questions in Weaknesses.

---

> ### Author Response · Authors · 2024-12-01
>
> We would like to express our sincere gratitude for the opportunity to revise our manuscript entitled “The Mutual Information Matrix in Hyperbolic Embedding and a Generalization Error Bound” and for the insightful comments. In response to the questions and concerns you have raised, we provide some answers here. In a new version we have submitted, we have updated some parts of the article's description:
> 1. Intuitively, a word $ w $ can be represented by a vector composed of the mutual information with all contexts $ c $ in $ V_c $, with a length of $ |V_c| $. This vector can effectively represent the information contained in a word. When two words have similar mutual information vectors with respect to context $ c $, it indicates a high degree of interchangeability in different contexts, which also suggests similarity in meaning. The dimension of the PMI matrix reflects the richness of information represented by different words through mutual information. However, since the number of contexts in $ V_c $ increases exponentially with the size of the chosen context window, if a lower-dimensional embedding of $ w $ can recover the elements of the mutual information matrix $PMI( w, c )$ through some function $ d(w, c) $, it would compress the embedding of $ w $ from $ |V_c| $ dimensions to a lower dimension. By analyzing the optimal solution of the objective function of the SGNS method, we discovered the relationship between the distance $ d(w, c) $ in hyperbolic embeddings and the mutual information between words $ w $ and contexts $ c $, as discussed in Section 3.1. Furthermore, by dividing the encoded $ d(w, c) $ matrix from the PMI matrix into errors introduced by the hyperbolic space and estimation errors caused by the SGNS method, we have defined the error bounds for hyperbolic embedding. The equivalence you mentioned refers to the first theorem we derived, which is the relationship between the dimension of the Gramian matrix of Poincaré embeddings and the dimension of the Poincaré space, the impact of the chosen Poincaré space dimension on estimation error.
>
> 2. In our revised version, we have enhanced the description of our experiments and theorem analysis to aid in understanding the results. Our experiments are mainly divided into four parts. First, we demonstrate that hyperbolic embedding significantly outperforms Euclidean space in recovering the dimensionality of the PMI matrix, reflecting the compressive capability of hyperbolic embedding, known as the parsimony property. Second, we present the dimensionality of the Gramian matrix of hyperbolic embeddings, as shown in Theorem 1. We then analyze the impact of different sample sizes on the embedding training error, verifying the results of Theorem 2, which states that low-dimensional hyperbolic spaces require more samples for training to achieve the same loss as Euclidean spaces. Finally, we discuss the reasons why the computational time for hyperbolic embeddings is longer than for Euclidean spaces, due to the need for the RSGD method and geodesic distance calculations, both of which greatly increase the computational complexity of hyperbolic embedding methods.
>
> 3. Taking your suggestions into account, in our revised version, we have added more detailed notes to our tables to explain their content and replaced some tables with line charts to more intuitively present our experimental results.

---

### Official Review · Reviewer_Wxoe · 2024-11-03

**Soundness:** 3
**Presentation:** 2
**Contribution:** 3
**Rating:** 5
**Confidence:** 3

**Summary:**

Hyperbolic embeddings were introduced in the literature as an alternative to the embeddings in Euclidean space. This paper provides an analysis of the skip-gram embedding model in hyperbolic space. The authors offer their take on many dimensions of the hyperbolic embeddings, including their connection to the mutual information matrix, generalization capabilities (with theoretical proof), and required sample size/training stability. Theoretical results are further supported by empirical results on two datasets: Wordnet and THUNews.

**Strengths:**

- I strongly believe exploring the embedding spaces beyond Euclidean space is crucial for the field
- Theoretical and empirical results are provided
- Reflection on the advantages (low-dimensionality) and disadvantages (training instability, large sample size etc.)

**Weaknesses:**

-  Although it's crucial and interesting to explore various properties of hyperbolic embeddings, they do not exist in a vacuum, so it would be useful to see the performance of the embeddings on downstream tasks
- Provided experimental setup and results are hard to follow (see questions)

**Questions:**

__Questions__:
- Why choose 400 Euclidean dimensionality and 2 for Poincare?
- Table 1,2,3: I don't really understand the reported numbers (what is the distance function exactly in Table 1? What is the distance in Table 2?). I suggest you give an explicit interpretation of those numbers to make it clear to the reader
- There is a conclusion that training with hyperbolic embeddings takes more time and iterations. However, it's unclear from your experiments if Poincare space embeddings can achieve the same loss as Euclidean ones with a higher number of samples (or iterations) (Table 6 and Table 7) or it is still behind the Euclidean embeddings
- Lime 418: `Moreover, hyperbolic space requires more than 70,000 samples to achieve adequate training`: what is _adequate_? How do you define it?

__Writing__:
- Table 7 has the incorrect title. It's 400-dimensional Euclidean space, not Poincare space

---

> ### Author Response · Authors · 2024-12-01
>
> We would like to express our sincere gratitude for the opportunity to revise our manuscript entitled “The Mutual Information Matrix in Hyperbolic Embedding and a Generalization Error Bound” and for the insightful comments. In response to the questions and concerns you have raised, we provide some answers here. In a new version we have submitted, we have updated some parts of the article's description:
> 1.  In the second part of our experiments in Section 4.2, to verify the relationship between the sample size and generalization error as stated in Theorem 2, we compared a 400-dimensional Euclidean space with a 2-dimensional Poincaré space. The reason for this choice is that when the amount of data approaches a very large scale, the errors of both methods are relatively close. At this point, the errors predicted by Theorem 2 are very small, implying that the errors introduced by the space, as discussed in Theorem 1, are close. This allows for a better comparison of the impact of data quantity on the generalization error across different spaces. In our new revised version, we have updated and explained the reasons for this choice.
> 2. In our revised version, we have enhanced the description of our experiments and theorem analysis to aid in understanding the results. Our experiments are mainly divided into four parts. First, we demonstrate that hyperbolic embedding significantly outperforms Euclidean space in recovering the dimensionality of the PMI matrix, reflecting the compressive capability of hyperbolic embedding, known as the parsimony property. Second, we present the dimensionality of the Gramian matrix of hyperbolic embeddings, as shown in Theorem 1. We then analyze the impact of different sample sizes on the embedding training error, verifying the results of Theorem 2, which states that low-dimensional hyperbolic spaces require more samples for training to achieve the same loss as Euclidean spaces. Finally, we discuss the reasons why the computational time for hyperbolic embeddings is longer than for Euclidean spaces, due to the need for the RSGD method and geodesic distance calculations, both of which greatly increase the computational complexity of hyperbolic embedding methods.
> 3. In our new revised version, we have added an explanation for fully trained models, as observed in the second experiment of Section 4.2, where full training indicates the sample size at which the loss converges, meaning the loss no longer changes significantly with an increase in the sample size.
> 4. As described in point 2, we have revised and changed the presentation of the experimental results in tables and curve charts.

---

> > ### Comment · Reviewer_Wxoe · 2024-12-03
> > **Keep my score**
> >
> > Thank you for your reply. I would strongly advise you to highlight the changes in the manuscript next time. Otherwise it's extremely hard to track revisions during the short rebuttal period.
> >
> > After reading the author responses and other reviews, I decided to keep my score unchanged.
> >
> > However, I want to stress that I do see the value in the paper, and encourage authors to give it some more work.

---

### Official Review · Reviewer_kgTB · 2024-11-03

**Soundness:** 3
**Presentation:** 3
**Contribution:** 1
**Rating:** 3
**Confidence:** 4

**Summary:**

the paper proposed to replace the Euclidean embeddings learned in word2vec with Hyberpoblic embeddings specifically with Poincare geometry. The method is straightforward - rather than using dot-product, a Euclidean-space similarity measure, the submission measures the distance between two word vectors on a Poincare disk. However, the evaluation approach puzzles me.

**Strengths:**

1. It directly replaces the distance/similarity measure in learning word2vec, which makes the approach easy to conceptualize.
2. Under mild assumptions, the submission provides interesting generalization bounds.

**Weaknesses:**

It puzzles me that there are many simple 'real-world'-ish datasets for evaluating learned word embedding, but somehow, the submission doesn't provide any of them. IMO, the submission conducts the study as if the problem is orthogonal to NLP.

1. Having an understanding of the sample complexity and how the error bound of the estimation depends on the sample complexity is generally informative, however, in recent years, we have found ourselves in a wacky situation that, for a model to generalize, the training loss just needs to be small, but it doesn't need to be very small, because many plateaus in the loss landscape provide models with good generalization, thus, having a theoretical understanding of the loss function or the error bound becomes somehow outdated.

2. A crucial aspect or consideration of learning on massive corpora is the complexity of the algorithm itself, which the submission doesn't mention.

3. The submission didn't use common datasets for learning word embeddings, nor does it provide any evaluation on common benchmarks, e.g. SimEval or SentEval.

**Questions:**

n/a

---

> ### Author Response · Authors · 2024-12-01
>
> We would like to express our sincere gratitude for the opportunity to revise our manuscript entitled “The Mutual Information Matrix in Hyperbolic Embedding and a Generalization Error Bound” and for the insightful comments. In response to the questions and concerns you have raised, we provide some answers here. In a new version we have submitted, we have updated some parts of the article's description:
> 1.  In our theoretical analysis, we chose a context window length of 1, hence $ V_c $ is the same as $ V_w $, and the samples used are in the form of word pairs. The WordNet dataset, being in the form of word pairs and featuring hierarchical relationships among nouns, is a common dataset in the study of Poincaré embedding methods. Therefore, our work also conducts comparative experiments on WordNet, and to avoid the influence of a single dataset, we have extended our tests to the newer and larger THUNews dataset. Additionally, due to the limitations of our experimental platform, we have not conducted training on larger datasets such as the Google News Corp.
>
> 2. Second, the bounds we propose in this paper include the bounds on generalization error. Furthermore, to minimize generalization error during training, we employ regularization methods to prevent overfitting, which can be partly attributed to an insufficient amount of training samples. For general word embedding, the complexity of word distribution in natural language often requires an immense amount of data to fit the distribution. To avoid overfitting when fitting the model on insufficient datasets, we do not want the training error to be too small. In this paper, we find that the mutual information vector between a word and its context can effectively represent the information of the word. However, since the context grows exponentially with the increase of the context window in the model, we aim to recover the word-context mutual information matrix through a lower-dimensional embedding as an embedding of the information between these words. Therefore, we focus on the embedding capability of these two methods for a specific PMI matrix on a given word dataset. We are more concerned with whether the resulting word embedding can well reflect the current co-occurrence probability distribution between the word and its context. Moreover, through our theory, we also understand that compared to Euclidean space, hyperbolic embedding methods require a larger sample size for training, thus more prone to overfitting on the same training set.
>
> 3. In the third part section 4.2, we discuss the complexity of the experimental methods. The dimensionality reduction capability of hyperbolic embedding requires more samples to achieve training errors close to those of Euclidean space. At the same time, due to the more complex calculations involved in using the RSGD method and hyperbolic geodesic distances, the algorithmic complexity of hyperbolic embedding methods is also more complex compared to Euclidean space.
>
> 4. Further, we added correlation tests for the WordSim353 dataset. Due to the limitations of the machine we used for embedding training, the Pearson correlation coefficients we obtained on the WordSim353 dataset are relatively low compared to the state of the art.. However, considering that the main goal of our study is to compare and understand the impact of the embedding space on the embeddings, the results we obtained still confirm our theoretical findings. The results we obtained are shown in the table below:
> | space      | dim | corre   | loss    |
> |------------|-----|---------|---------|
> | euclidean  | 50  | 0.9997  | 1.6177  |
> | euclidean  | 100 | 0.1132  | 1.5586  |
> | euclidean  | 150 | 0.1187 | 1.5164  |
> | euclidean  | 200 | 0.1228 | 1.4988  |
> | poincare   | 2   | 0.1189  | 0.9214  |
> | poincare   | 4   | 0.1206  | 0.7809  |
> | poincare   | 6   | 0.1302  | 0.7413  |
> | poincare   | 8   | 0.1040  | 0.7915  |
> From the experimental results, we can observe that the loss we obtained is positively correlated with the Pearson correlation coefficient. When we increase the dimensions of Euclidean and Poincare spaces, the training loss decreases, accompanied by an increase in the Pearson correlation coefficient. For the Poincare space, compared to the Euclidean space, a high Pearson correlation coefficient can be achieved with a relatively low number of dimensions. However, when the dimension is equal to 8, due to the insufficiency of training samples, the training of the Poincare embedding is not adequate, resulting in a lower Pearson correlation.

---

### Official Review · Reviewer_pY2V · 2024-11-04

**Soundness:** 2
**Presentation:** 2
**Contribution:** 2
**Rating:** 3
**Confidence:** 2

**Summary:**

This paper provides both theoretical and empirical analysis of Skip-Gram Negative-Sampling (SGNS) embeddings in hyperbolic space. While SGNS traditionally embeds words and contexts in Euclidean space (Word2Vec), the authors extend this approach to hyperbolic space using  Poincaré embeddings. Two types of errors are used to evaluate the embeddings: spatial error, which is influenced by the dimensions and structure of hyperbolic space, and generalization error, which measures the relationship between embedding error and sample size across different spaces. An empirical study of hyperbolic embeddings is conducted on WordNet and THUNews.

The authors investigate how hyperbolic distance relates to mutual information, deriving bounds on both spatial and generalization errors. Furthermore, they demonstrate that the distance, d(w,c), between w and c corresponds to the mutual information between w and c in a hyperbolic space. This finding helps to motivate the use of Poincaré embeddings.

**Strengths:**

The paper provides novel insights by studying the mutual information matrix in Skip-Gram Negative-sampling (SGNS) embeddings in hyperbolic space. In particular, demonstrating that distance in hyperbolic embeddings obtained by using SGNS equates to mutual information is an interesting finding that can motivate the use and further study of Poincaré embeddings in NLP. Additionally, the empirical result that hyperbolic embeddings are more unstable during training than their Euclidean counterpart and that more samples are needed to reduce training error can help guide further works in training hyperbolic embeddings.

**Weaknesses:**

Overall, the paper is extremely dense and difficult to follow because it provides little motivation or intuition for mathematical notation.

I understand that one of the paper's main contributions is to provide a detailed mathematical analysis of the mutual information matrix in hyperbolic embeddings. Still, some detail is unnecessary in the main body of the paper and hinders the reader's ability to read the paper. For example, results such as those in section 3.1 that use straightforward algebraic computations to show that distance approximates mutual information should be moved to the appendix.

While the paper provides some nice theoretical insights, the methods used for the evaluation of hyperbolic embeddings with Skip-Gram Negative-Sampling are not robust.  Using the rank of the restored point-wise mutual information matrix as the sole metric to compare Euclidean hyperbolic embeddings is not particularly interesting. Investigating the performance of hyperbolic embeddings on word similarity tasks, e.g., WordSim-353 or SimLex999, would provide a meaningful quantitative comparison of using embeddings based in different spaces and help motivate the study of static, hyperbolic word embeddings. Further, comparing the performance of classification models that use standard Word2Vec embeddings and hyperbolic Skip-Gram Negative sampling embeddings would provide a much stronger motivation for the paper.

**Questions:**

1. 263-264: I don’t understand the reasons for setting $V_{w} = V_{c} = V$. Can you elaborate more on why this setting is used? If it is common practice, there should be some citations.

2. It would be helpful to provide a clear definition of parsimony in Section 3.2

---

> ### Author Response · Authors · 2024-12-01
>
> We would like to express our sincere gratitude for the opportunity to revise our manuscript entitled “The Mutual Information Matrix in Hyperbolic Embedding and a Generalization Error Bound” and for the insightful comments. In response to the questions and concerns you have raised, we provide some answers here. In a new version we have submitted, we have updated some parts of the article's description:
> 1. We have removed the algebraic computation from Section 3.1 of the main text and moved them to the appendix, while also adding more motivation and intuitive understanding for the substitution of the PMI matrix. Furthermore, we have added motivation and intuitive understanding for Section 3.2 as well.
> 2. The reason we choose the order of the PMI matrix as an indicator is that in the PMI matrix, the mutual information between different words and the context forms vectors that serve as an embedding for the words. For words with similar mutual information vectors, it indicates that they contribute similar information in different contexts, thus showing a high degree of substitutability. However, these vectors are usually of high dimension, so we need to reproduce them through a lower-dimensional representation.  As for PMI matrices of higher rank, it indicates a greater degree of differentiation between words, which also preserves more information between words.
> 3. Regarding the first question you raised, we have added the reasons for choosing $ V = V_w = V_c $. Since the size of $ V_c $ grows exponentially with the size of the context window, and the dimension of the $ V_w \times V_c $ matrix is at most $ V_w $, we have chosen the context window to be 1, meaning the context vocabulary is the same as the word vocabulary. At this time, when we use the SGNS training method, we are using a word pair-shaped dataset. Moreover, the distance matrix at this time is a square matrix, which is also easier for following theoretical analysis. When the context window decreases, it may become difficult to distinguish the mutual information vectors corresponding to different words, making it hard to differentiate between word vectors, thus reducing the rank of the PMI matrix.
> 4. For the second question you raised, we have clearly defined the concept of parsimony at the beginning of Section 3.2: Compared to Euclidean space, hyperbolic space can compress the dimensions of the embeddings into a lower-dimensional space.
> 5. Further, we added correlation tests for the WordSim353 dataset. Due to the limitations of the machine we used for embedding training, the Pearson correlation coefficients we obtained on the WordSim353 dataset are relatively low compared to the state of the art.. However, considering that the main goal of our study is to compare and understand the impact of the embedding space on the embeddings, the results we obtained still confirm our theoretical findings. The results we obtained are shown in the table below:
> | space      | dim | corre   | loss    |
> |------------|-----|---------|---------|
> | euclidean  | 50  | 0.9997  | 1.6177  |
> | euclidean  | 100 | 0.1132  | 1.5586  |
> | euclidean  | 150 | 0.1187 | 1.5164  |
> | euclidean  | 200 | 0.1228 | 1.4988  |
> | poincare   | 2   | 0.1189  | 0.9214  |
> | poincare   | 4   | 0.1206  | 0.7809  |
> | poincare   | 6   | 0.1302  | 0.7413  |
> | poincare   | 8   | 0.1040  | 0.7915  |
> From the experimental results, we can observe that the loss we obtained is positively correlated with the Pearson correlation coefficient. When we increase the dimensions of Euclidean and Poincare spaces, the training loss decreases, accompanied by an increase in the Pearson correlation coefficient. For the Poincare space, compared to the Euclidean space, a high Pearson correlation coefficient can be achieved with a relatively low number of dimensions. However, when the dimension is equal to 8, due to the insufficiency of training samples, the training of the Poincare embedding is not adequate, resulting in a lower Pearson correlation.

---

### Meta-Review · Area_Chair_RDpK · 2024-12-20

**Metareview:**

This paper is concerned with performing Word2Vec-style embeddings in hyperbolic space. The chosen model of hyperbolic space is the Poincare disk. The authors derive some theoretical results that generalize the Euclidean setting and perform some experiments for the hyperbolic version of the algorithm.

One strength is that it is always interesting to know how non-Euclidean variants of Euclidean methods work. This paper joins a line of work performing this task.

There are also a few big weaknesses for this paper. First, it is written in a style that makes it difficult to grasp, even for those who are fairly familiar with these topics. Second, the contribution here appears limited. Hyperbolic word embeddings are not a new topic (see for example Tifrea et al ’18). The authors can argue that for the particular combination of Word2Vec-type approach with hyperbolic space there is novelty, but it’s not clear what the gain is; the experimental findings are similar.

The second weakness is that the experiments are very limited. There are only a couple of datasets. These weaknesses were consistently agreed upon by the reviewers. As a result, this paper is not quite ready for acceptance, but could be the start of a solid work for down the line.

**Additional Comments On Reviewer Discussion:**

Reviewers focused on two aspects: the structure of the presentation and the clarity of the writing, along with the limited experimental results. The authors addressed some of the clarity concerns, but the paper could definitely use another round of improvements.

---

### Decision · Program_Chairs · 2025-01-22

Reject